# Development of AlissAID system targeting GFP or mCherry fusion protein

**Yoshitaka Ogawa, Kohei Nishimura * , Keisuke Obara , Takumi Kamura ***

Department of Biological Science, Division of Natural Science, Graduate School of Science, Nagoya University, Nagoya, Japan

☯ These authors contributed equally to this work.

* nishimura.kohei.x8@f.mail.nagoya-u.ac.jp (KN); kamura.takumi.k1@f.mail.nagoya-u.ac.jp (TK)

**Data Availability Statement:** All relevant data are within the paper and its Supporting Information files. Plasmids are available from Addgene (https://www.addgene.org/Takumi_Kamura/) at accession

## Abstract

Conditional control of target proteins using the auxin-inducible degron (AID) system provides a powerful tool for investigating protein function in eukaryotes. Here, we established an Affinity-linker based super-sensitive auxin-inducible degron (AlissAID) system in budding yeast by using a single domain antibody (a nanobody). In this system, target proteins fused with GFP or mCherry were degraded depending on a synthetic auxin, 5-Adamantyl-IAA (5-Ad-IAA). In AlissAID system, nanomolar concentration of 5-Ad-IAA induces target degradation, thus minimizing the side effects from chemical compounds. In addition, in AlissAID system, we observed few basal degradations which was observed in other AID systems including ssAID system. Furthermore, AlissAID based conditional knockdown cell lines are easily generated by using budding yeast GFP Clone Collection. Target protein, which has antigen recognition sites exposed in cytosol or nucleus, can be degraded by the AlissAID system. From these advantages, the AlissAID system would be an ideal protein-knockdown system in budding yeast cells.

## Author summary

Budding yeast has been widely studied as a genetic model organism for eukaryotic species. Targeted degradation systems, in which conditional-control of target protein degradation, are used to examine protein function. The auxin-inducible degron (AID) system is one such system that rapidly disrupts target proteins in an auxin (IAA)-dependent manner. In this system, high concentrations of auxin and AID-tagging to the target proteins are necessary for degradation. Here, we developed the Affinity-linker based super-sensitive auxin-inducible degron (AlissAID) system, which enables to degrade target proteins with nM concentration of a synthetic auxin, 5-Adamantyl-IAA (5-Ad-IAA). In this system, we can use popular fluorescent proteins GFP or mCherry as degradation tags and avoid AID-tagging to the target endogenous gene. As GFP Clone Collection is commercially available in budding yeast, it is easy to generate conditional mutants in AlissAID system. AlissAID system would be a powerful tool of genetic analysis in eukaryotic species including budding yeast.

number #198412, #198413, #198414, #198415, #198416, and #198417.

**Funding:** KN received support from the Japan Society for the Promotion of Science KAKENHI Grant Numbers, JP19K06611, JP20K21423 and JP22K05558. KN also received support from Kato Memorial Bioscience Foundation and Institute for Fermentation, Osaka. KO received support from the Japan Society for the Promotion of Science KAKENHI Grant Number, JP22K06141 and from Institute for Fermentation, Osaka. TK received support from the Japan Society for the Promotion of Science KAKENHI Grant Number, JP20H03208. The funders had no role in study design, data collection and analysis, decision to publish, or preparation of the manuscript.

**Competing interests:** The authors declare no competing financial or non-financial interests.

## Introduction

Conditional target-protein knockdown systems are used to analyze gene function. In particular, the auxin-inducible degron (AID) system [1–5] has been widely used to study protein-knockdown in various types of eukaryotes. In this method, AID-tagged target proteins are polyubiquitinated by SCF-OsTIR1 (Os referring to *Oryza sativa*), in an auxin-dependent manner (Fig 1A). The polyubiquitinated proteins are then recognized and degraded by the 26S proteasome [6–9]. The conventional AID system requires over 100 μM of indole-3-acetic acid (IAA) for target protein degradation, leading to cytotoxicity in some organisms [10].

To reduce the cytotoxicity caused by high IAA concentrations, super-sensitive AID (ssAID) [11] and auxin-inducible degron 2 (AID2) [12] systems have been developed in mammalian cell lines and applied fission yeast [13,14]. Both systems utilize mutated OsTIR1 (the F74A mutant in ssAID and the F74G mutant in AID2) and enable target protein degradation by lowering the concentration of synthetic auxins (Fig 1B). In the ssAID system with the OsTIR1$^{F74A}$, a nanomolar concentration of 5-adamantyl-IAA (5-Ad-IAA) is sufficient for target-protein degradation via polyubiquitination. These improved AID systems could avoid cytotoxicity by reducing the concentration of inducer required for target protein degradation.

In these AID systems, an AID-tag must be fused with the target protein intended for degradation. Fusion with the AID-tag might affect target-protein stability and function. In some cases, co-expression with TIR1 causes weak degradation of the AID-tagged target without induction, which is termed basal degradation [15–17]. One possible cause of basal degradation

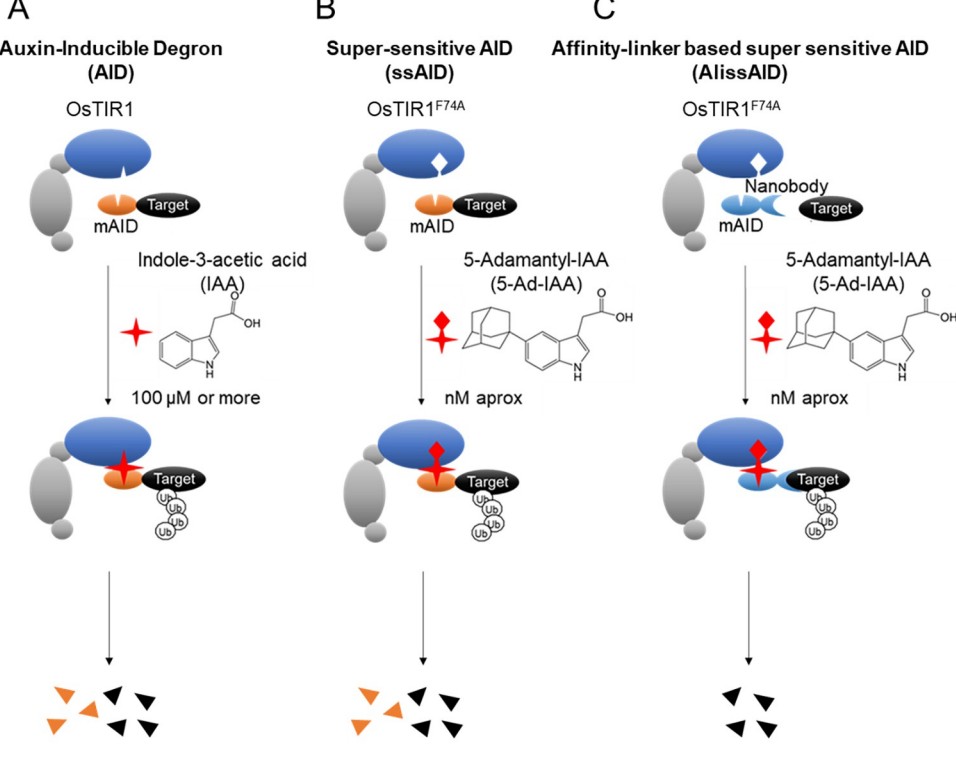

**Fig 1. Schematic illustration of the Auxin-Inducible Degron (AID) systems.** (A) the conventional Auxin-Inducible Degron (AID) system, (B) super-sensitive AID (ssAID), and (C) Affinity-linker based super sensitive AID (AlissAID) systems.

is that small amounts of IAA (or similar chemicals) induce degradation of target proteins. IAA is a metabolite of tryptophan, and similar metabolites are present in non-plant organisms as well [18,19]. Owing to basal degradation, it is sometimes difficult to generate AID-based conditional knockdown cell lines for essential proteins. Some improved AID systems (including ssAID and AID2) showing limited basal degradation have been developed in mammalian cell lines [11,12,16,17] and fission yeast [13]. However, IAA induced a weak interaction between OsTIR1$^{F74A}$ and AID-tag in the yeast two hybrid assay [11].Therefor, it is possible that basal degradation is an issue to be solved in AID system of budding yeast.

Single-peptide antibody nanobodies [20,21] are useful for protein recognition in target protein degradation systems. Several studies have reported that fusion proteins of E3 ligase and the nanobody can induce degradation of target proteins recognized by nanobodies. Such targeted degradation systems have been developed in studies using *Drosophila* [22] and have been applied to human cells [23–25]. The VHHGFP4 nanobody, which binds to green fluorescent protein (GFP) has been applied to the conventional AID system [26]. GFP fusion protein was degraded in cultured human and zebrafish cells expressing both wild-type OsTIR1 (OsTIR1$^{WT}$) and a chimeric protein-fused AID-tag containing VHHGFP4. In this chimeric protein, the lysine residues have been replaced by arginine residues; as a result, OsTIR1 can ubiquitinate the target protein bound to the chimeric protein without ubiquitinating the chimeric protein. The GFP-tag is a stable tag that is widely used for target-protein visualization, and GFP-tagged lines have been generated for various species. Thus, A GFP fusion-protein-targeting AID system represents a powerful molecular research tool for eukaryotes.

The budding yeast *Saccharomyces cerevisiae* is a model organism for eukaryotic cells. In budding yeast, the GFP-tag can be easily fused with an endogenous protein via genetic manipulation. A Yeast GFP Clone Collection [27] (Thermo Fisher scientific), with a GFP-tagged Open Read Frame (ORF) at its chromosomal locus, that contains 4,159 GFP-tagged ORFs, comprising 75% of the yeast proteome, is currently available. We combined a nanobody system with our ssAID system to develop an Affinity-linker based super-sensitive auxin-inducible degron (AlissAID) system in budding yeast (Fig 1C). AlissAID-based conditional-knockdown cells can be generated by the expression both OsTIR1$^{F74A}$ and a chimeric protein AID-tag fused with the nanobody in the GFP-tagged strain. This system utilizes 5-Ad-IAA as a degradation inducer and induces target-protein degradation at nanomolar concentrations of inducer. We also found that it can be applied to the degradation of mCherry fusion proteins, which is recognized with LaM2 or LaM4 nanobodies. The AlissAID system is therefore useful to analyze target-protein physiological function in budding yeast cells.

## Results

### Comparison of AlissAID system with other AID systems

The AID system enables rapid degradation of AID-tagged target proteins within cells via the natural auxin IAA, requiring approximately 500 μM of IAA for target protein degradation (Fig 1A) [1]. In contrast, the ssAID system, which uses the OsTIR1$^{F74A}$ mutant and an artificial auxin, 5-Ad-IAA, allows target-protein degradation at a nanomolar concentration of 5-Ad-IAA (Fig 1B) [11]. Therefore, by using the ssAID system, it may be possible to avoid the side effects of high concentrations of degradation inducers. In both AID systems, AID-tags are necessary for target protein degradation. However, AID-tags can affect target-protein stability or function. Here, we attempted to eliminate these effects by using nanobodies for target-protein recognition (Fig 1C). In the AlissAID system, the AID-tagged nanobody binds to the target protein. The AID-tag was recruited to OsTIR1$^{F74A}$ in a 5-Ad-IAA-dependent manner. All the lysine residues of the AID-tagged nanobody were replaced with arginine, preventing it from

being ubiquitinated by the SCF complex and OsTIR1$^{F74A}$. Instead, the target protein is ubiquitinated and degraded by the 26S proteasome.

## OsTIR1$^{F74A}$ is suitable for the AlissAID system

The use of OsTIR1$^{WT}$ with nanobodies can degrade GFP fusion protein in mammalian cells and zebrafish embryos [26]. Additionally, in mammalian cell lines, certain OsTIR1 mutants effectively degrade target proteins [11]. To compare the degradation effectiveness of OsTIR1 mutants, we generated an Mcm4-GFP degron strain by expressing OsTIR1 (wild-type [WT], and mutants F74A, F74S, or F74C) and a minimal AID-tag fused with an anti-GFP nanobody (mAID-Nb) in which all lysine residues were replaced by arginine residues, under the control of the constitutive ADH1 promoter. Mcm4-GFP was severely degraded in the presence of 5 μM 5-Ad-IAA in OsTIR1$^{F74A}$, while it was only slightly degraded in the OsTIR1$^{F74S}$ and OsTIR1$^{F74C}$ strains. However, Mcm4-GFP was barely degraded following the addition of 500 μM IAA in the OsTIR1$^{WT}$ strain (Fig 2A and 2B). A similar result was obtained via serial dilution spotting analysis. OsTIR1$^{F74A}$-expressing cells exhibited severe growth defects, even in the presence of 50 nM 5-Ad-IAA. OsTIR1$^{F74S}$- and OsTIR1$^{F74C}$-expressing cells required 500 nM 5-Ad-IAA to generate growth defects, while OsTIR1$^{WT}$-expressing cells required 500 μM IAA (Fig 2C). Furthermore, minimal target degradation was observed in AlissAID strains with OsTIR1$^{F74G}$, which has been reported to reduce basal degradation. (S1A and S1B Fig). These results indicate that OsTIR1$^{F74A}$ is the suitable OsTIR1 mutant for use in the AlissAID system in budding yeast.

## GFP fusion protein is rapidly degraded in the OsTIR1$^{F74A}$-and-mAID-Nb expressing cell lines

We generated AlissAID strains toward some essential genes in budding yeast, *S. cerevisiae*. We expressed both OsTIR1$^{F74A}$ and mAID-Nb in the W303-1a background [28]. We added a GFP-tag to the essential proteins Mcm4, Ask1, and Neo1. Immunoblots showed that these target proteins were degraded within 45 min after the addition of 5-Ad-IAA (Fig 3A). The target protein degradation was also confirmed with a immunoblot by using an antibody which recognized endogenous Mcm4 (S2A Fig). Next, we compared the degradation efficiency of target proteins with different expression levels. The degradation efficiencies of Mcm4 (middle expression), Ask1 (low expression) and Neo1 (high expression) are almost similar in this AlissAID system (S2B and S2C Fig). Fluorescent microscopy revealed that, following 5-Ad-IAA addition, Mcm4-GFP (localized in the nucleus and cytoplasm), Ask1-GFP (localized to the nucleus), and Neo1-GFP (localized at the membrane) GFP signals were reduced (Fig 3B). For these cell lines expressing OsTIR1$^{F74A}$ and mAID-Nb, serial dilution spotting to confirm growth-defect phenotypes revealed severe growth defects, even in the presence of 50 nM 5-Ad-IAA (Fig 3C). AlissAID strains exhibited 5-Ad-IAA dependent phonotypes at 20˚C, 24˚C, and 37˚C, indicating that AlissAID system works under the wide range of temperature conditions (Fig 3D). Mcm4 and Ask1 have been reported to be essential for the progression of DNA replication in S phase and chromosomal segregation in G2/M phases, respectively [29,30]. FACS analysis showed that Mcm4-GFP cells and Ask1-GFP cells started to arrest at S phase and at G2/M phase 1 h after the addition of 5-Ad-IAA, respectively (S2D and S2E Fig). These results indicate that the AlissAID system works well in budding yeast by enabling the degradation of nuclear, cytosolic, and membrane proteins at low concentrations of 5-Ad-IAA.

## AlissAID system reduces target-protein basal degradation

There are questions about the extent to which basal degradation occurs in AlissAID system of budding yeast. To approach this, we evaluated basal degradation in the ssAID and AlissAID

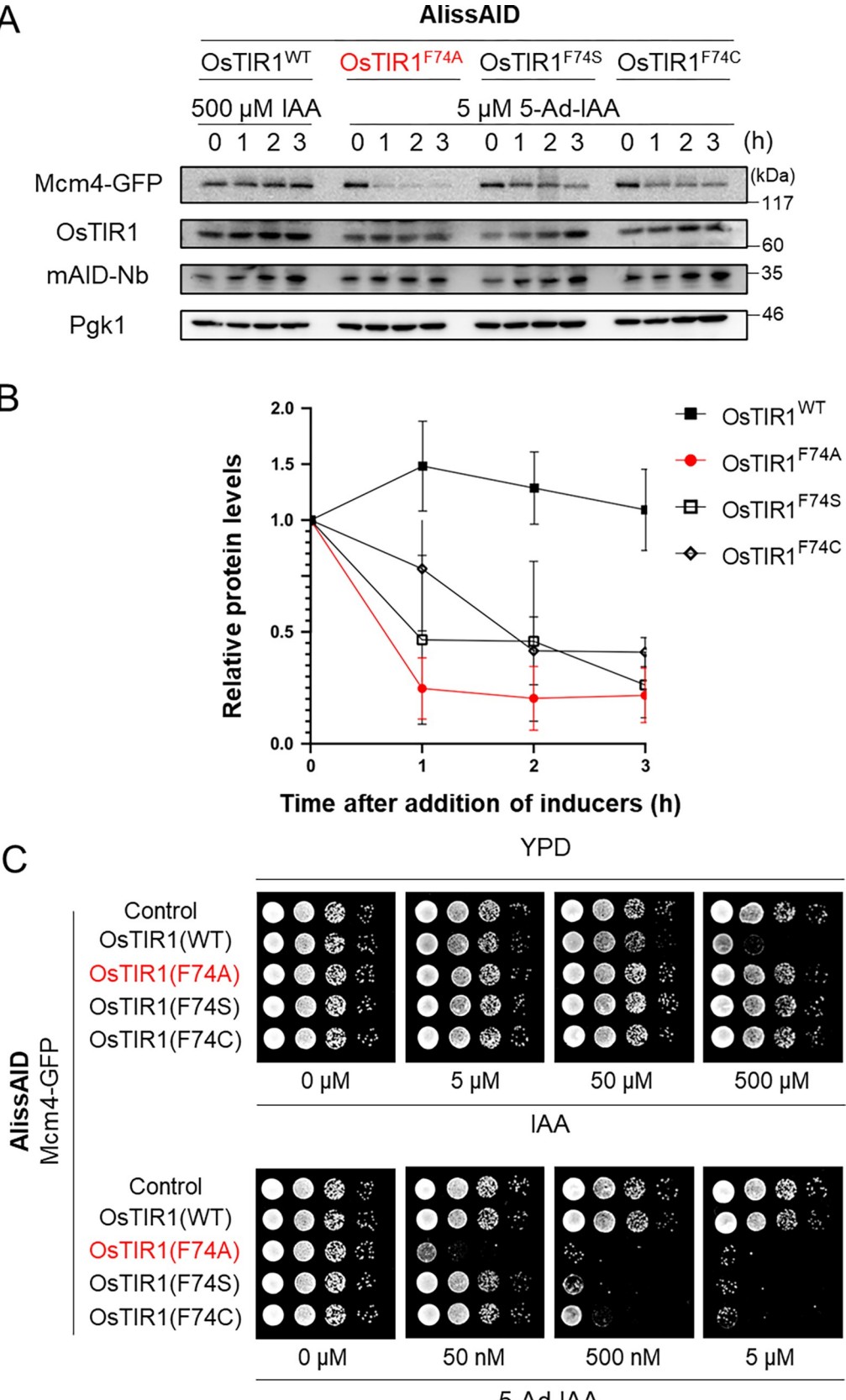

**Fig 2. The OsTIR1$^{F74A}$ mutant most efficiently removes target proteins.** (A) Immunoblots of AlissAID strains with various OsTIR1 mutants, showing the 5-Ad-IAA-triggered degradation of a target protein (Mcm4-GFP). All strains were treated with 500 μM IAA or 5 μM 5-Ad-IAA and sampled at the indicated time points. Pgk1 was used as a loading control. (B) After normalizing with the loading control Pgk1, the relative Mcm4-GFP levels to those at time zero are shown as means ± SD (n = 3 biological replicates).(C) Serial dilution spotting assay of each AlissAID strain on YPD medium with the indicated concentrations of IAA or 5-Ad-IAA. All strains that had an essential protein (Mcm4-GFP) as a target protein were grown for 24 h at 30°C.

systems of budding yeast. We generated two strains: (i) an Neo1-ssAID strain based on OsTIR1$^{F74A}$-expressing cell, in which C terminus of Neo1 was tagged by mAID-GFP and (ii) an Neo1-AlissAID strain based on OsTIR1$^{F74A}$-and-mAID-Nb-expressing cell, in which C terminus of Neo1 was tagged by GFP. We observed a substantial reduction in Neo1 protein levels even in the absence of 5-Ad-IAA in the ssAID system (Fig 4A).

This reduction would be due to the basal degradation. To confirm it, we further generated the control strain lacking the OsTIR1$^{F74A}$-expression. Neo1 protein was degraded with 5-Ad-IAA treatment in AlissAID and ssAID strains but not in the control strains (Fig 4B and 4C). Importantly, even in the absence of 5-Ad-IAA, the expression levels of Neo1-mAID-GFP were reduced in ssAID strain (Fig 4C). On the other hand, the expression levels of Neo1-GFP in the AlissAID strain did not change in the absence of 5-Ad-IAA (Fig 4B). We obtained similar results with Mcm4 and Ask1 in AlissAID strain (S3 Fig). To confirm the effect of basal degradation in ssAID on the function of Neo1 (Neomycin resistant 1) protein in budding yeast, we examined the G418 (neomycin analogue) sensitivity of the AlissAID and ssAID strains in the absence of 5-Ad-IAA by serial dilution spotting assay. The ssAID strain exhibited defective growth on G418-containing medium, whereas the AlissAID strain did not show any growth defects (Fig 4D). These results are corelated with those of basal degradation in the ssAID and AlissAID strains. This means that the AlissAID system shows milder basal degradation than the ssAID system in budding yeast.

## Use of a yeast GFP clone collection in conjunction with the AlissAID system enables easy generation of a degron cell line

There is the budding yeast GFP Clone Collection [27] in the BY4741 [31] background. The GFP gene is integrated into the C-terminus of the endogenous gene via homologous recombination, and the GFP fusion protein is expressed under the endogenous promoter. This fusion collection is useful for generating GFP-degron cells via the AlissAID system to target budding yeast proteins (Fig 5A). Therefore, we constructed a plasmid by combining OsTIR1$^{F74A}$ and mAID-Nb under the control of the *ADH1* promoter on pRS316 [32], with OsTIR1$^{F74A}$ and mAID-Nb connected by self-cleavage T2A sequence (Fig 5B). First, we generated the Neo1-degron strain by transforming the plasmid into Neo1-GFP. Neo1-GFP protein was rapidly degraded after the addition of 5-Ad-IAA (Fig 5C). Subsequently, we generated an AlissAID strain with various essential proteins by transforming this plasmid into GFP clones. While all the cells grew well in the absence of 5-Ad-IAA, severe growth defects were observed in the presence of 500 nM 5-Ad-IAA (Fig 5D), and GFP signals were reduced in the presence of 5-Ad-IAA (S4A Fig). These results indicate that the combined application of a GFP clone collection and the AlissAID system enables easy generation of degron cell lines. This combined system offers great potential for studying protein function in budding yeast.

We next examined GFP-tagged proteins localized in different compartments; Sam35 in mitochondria and Dpm1 in endoplasmic reticulum (ER). Sam35 is a component of SAM complex located in the mitochondrial outer membrane, while Dpm1 is a transmembrane protein of the ER membrane with its C-terminus exposed to the ER lumen (S4B Fig). We constructed

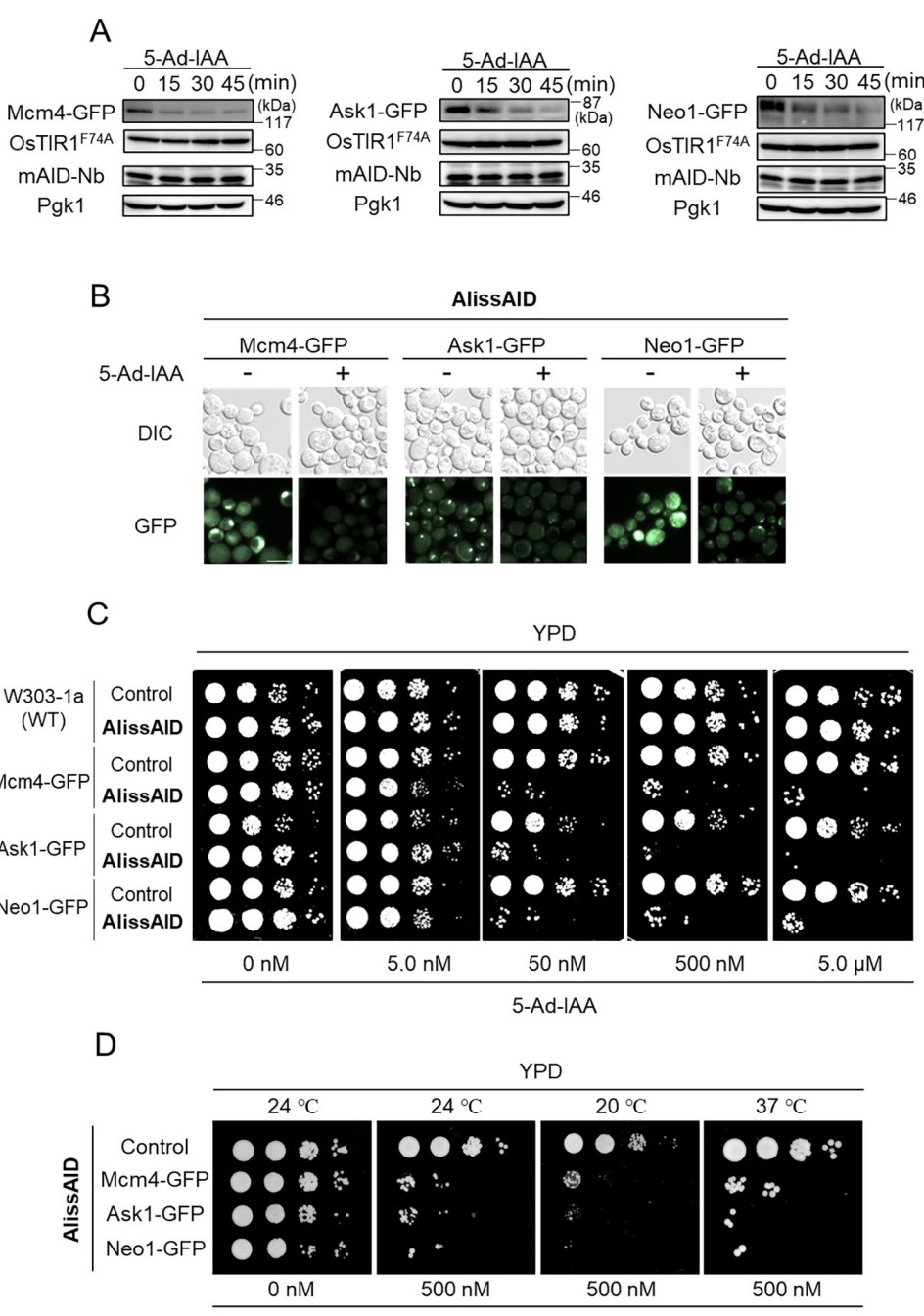

**Fig 3. Efficient degradation of GFP fusion proteins in OsTIR1$^{F74A}$ and mAID-Nb-expressing cells in the presence of 5-Ad-IAA.** (A) Immunoblotting analysis of GFP fusion proteins Mcm4, Ask1, and Neo1. Cells were treated with 5 μM 5-Ad-IAA and sampled at the indicated time points. Pgk1 was used as a loading control. (B) Fluorescence microscopy observations of Mcm4-, Ask1-, or Neo1-GFP. Cells are treated with or without 5.0 μM 5-Ad-IAA for 3 h. White bar, 5 μm. (C, D) Serial dilution spotting on YPD medium containing 5-Ad-IAA. Cells were grown for 30 h at 30°C (C), for 24h at 20°C, 24°C or 37°C (D).30°C.

the Sam35- and Dpm1-AlissAID strains by fusing GFP-tag at their C-termini. Immunoblots showed that Sam35-GFP was degraded, while Dpm1-GFP was not degraded in a 5-Ad-IAA-dependent manner (S4C Fig). Degradation of non-essential protein Sam35 caused no growth

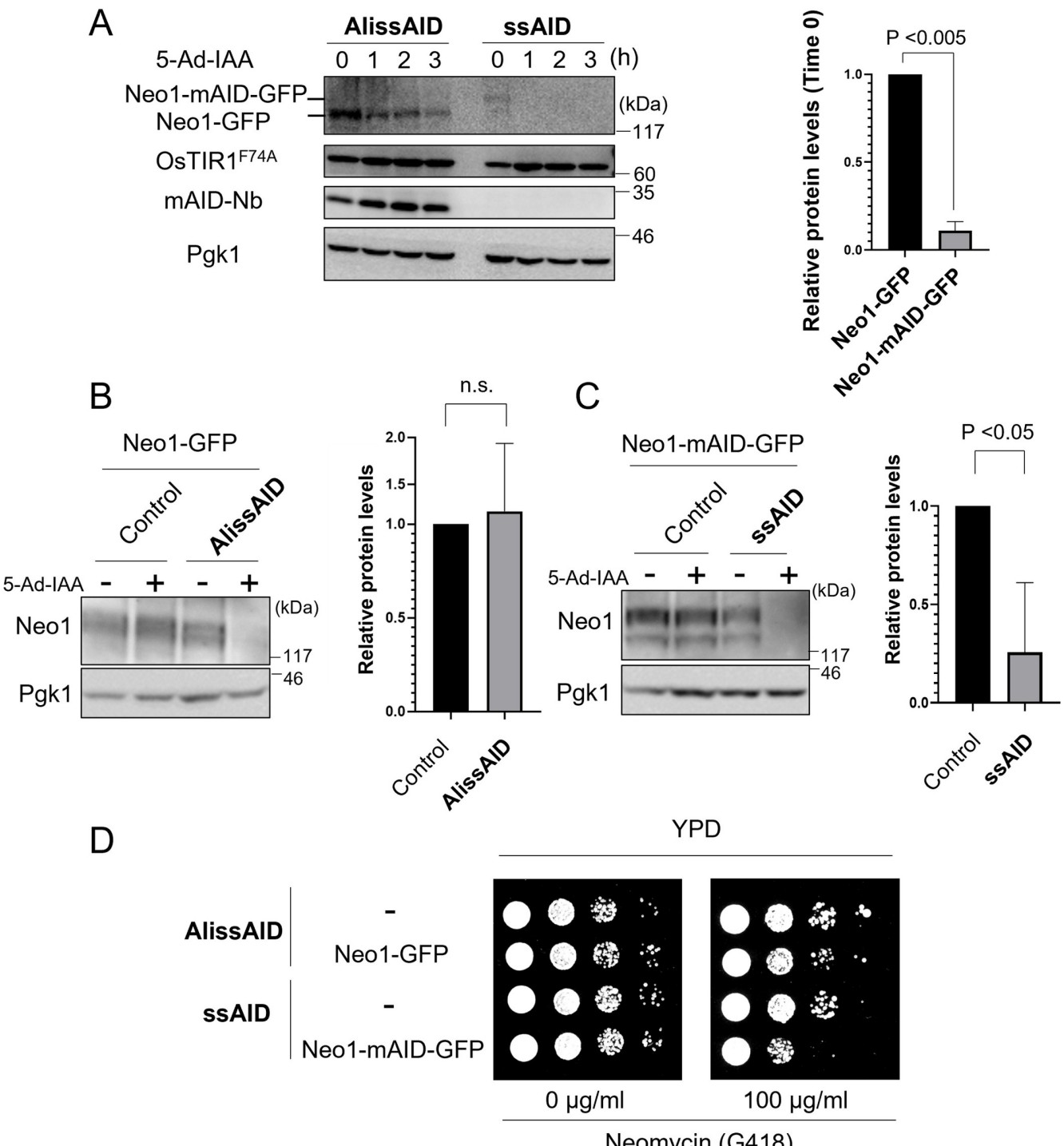

**Fig 4. The AlissAID system reduces basal degradation and side effects.** (A) Immunoblots of AlissAID and ssAID strains after applying 5-Ad-IAA, showing the quantitative changes of target proteins (GFP-tagged Neo1) (left). After normalizing with the loading control Pgk1, target protein levels at time zero were statistically analysed (right). Means ± SD (n = 3 biological replicates), p < 0.05; unpaired Student's *t*-test. (B, C) Immunoblots of AlissAID strain (B) and ssAID strain (C), both of which were treated with (+) or without (-) the 5-Ad-IAA for 3 h, showing the target protein levels (left). Control strains had their respective target protein (Neo1-GFP or Neo1-mAID-GFP), but not OsTIR1 F74A. After normalizing as in (A), target protein levels without 5-Ad-IAA are shown (right). Means ± SD (n = 3 biological replicates); n.s., nonsignificant; p < 0.05; unpaired Student's *t*-test. (D) Serial dilution spotting assay on YPD medium with or without neomycin (G418). AlissAID and ssAID strains were grown for 24 h or 54 h at 30˚C.

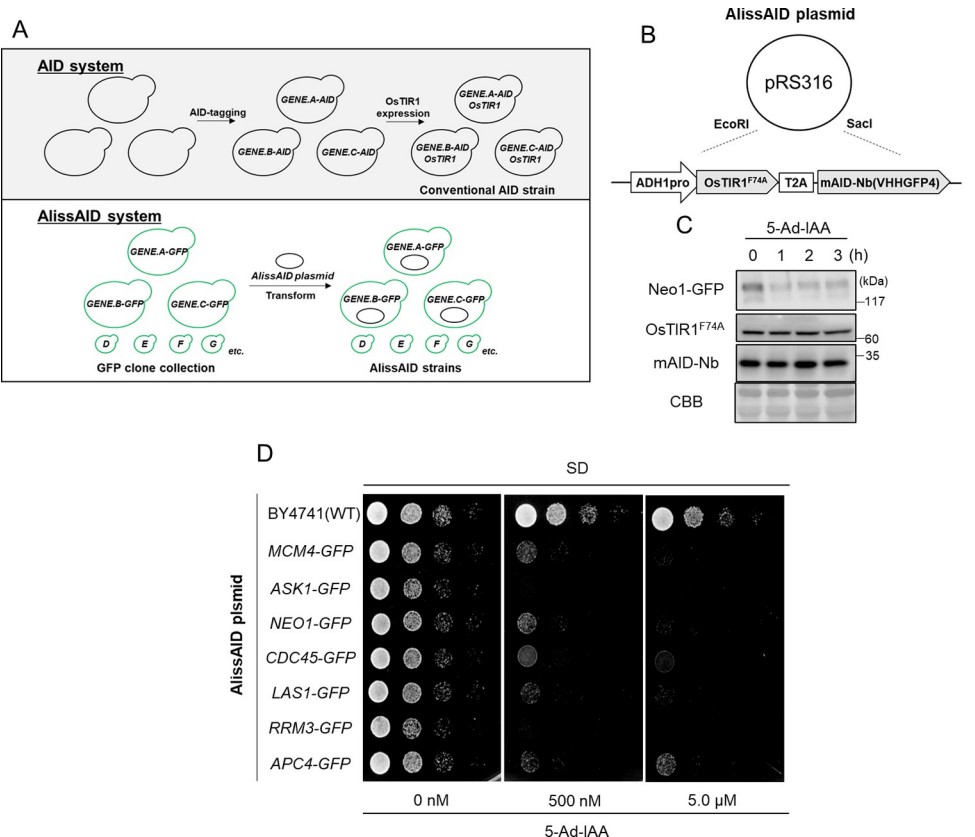

**Fig 5. Target-protein degradation and phenotyping in GFP Clone cells with single-plasmid coding OsTIR1$^{F74A}$ and mAID-Nb.** (A) Degron strain preparation of conventional AID system and AlissAID system. The AID system requires two steps of gene editing: AID-tagging of the target gene and expression of OsTIR1. The AlissAID system using the AlissAID plasmids and the GFP Clone Collection produces a variety of GFP-knockdown strains in a single-step transformation. (B) Overview of AlissAID plasmid. ADH1 promoter, OsTIR1$^{F74A}$, T2A, mAID-Nanobody sequences cloned between EcoRI, Sac1 site of pRS316 vector. (C) Immunoblotting of endogenous GFP-tagged Neo1 cells transformed with the plasmid. Cells were treated with 5 μM 5-Ad-IAA. (D) Serial dilution spotting on SD medium containing 5-Ad-IAA. Cells were grown for 24 h at 30°C.

defects on 5-Ad-IAA-containing medium as did non-degradation of Dpm1 (S4D Fig). Taken together, these results indicate that antigen recognition sites must be exposed in cytosol or nucleus in the AlissAID system, as in the conventional AID system [1].

## mCherry fusion proteins were degraded by the AlissAID system using anti-mCherry nanobody, LaM2 and LaM4

Similar to GFP, the red fluorescent protein mCherry is commonly used in budding yeast and some anti-mCherry nanobodies (named LaMs, [33]) are available. We selected three LaMs (LaM2, LaM4 and LaM8) that have different properties in binding sites and binding forces. We added mCherry tag to the C-terminus of the essential protein Ask1 via homologous recombination and expressed both OsTIR1$^{F74A}$ and mAID-LaM2 and LaM4 or LaM8 [33] in the W303-1a background (S5A Fig). These cells expressed the same levels of each mAID-Nb (LaM2, LaM4, or LaM8) (S5B Fig). Immunoblotting analysis revealed that Ask1-mCherry was highly degraded in cell lines with mAID-LaM2 and weakly degraded in cell lines with mAI-D-LaM4, but not in cell lines with mAID-LaM8 (Fig 6A). Further, after the addition of 5-Ad-

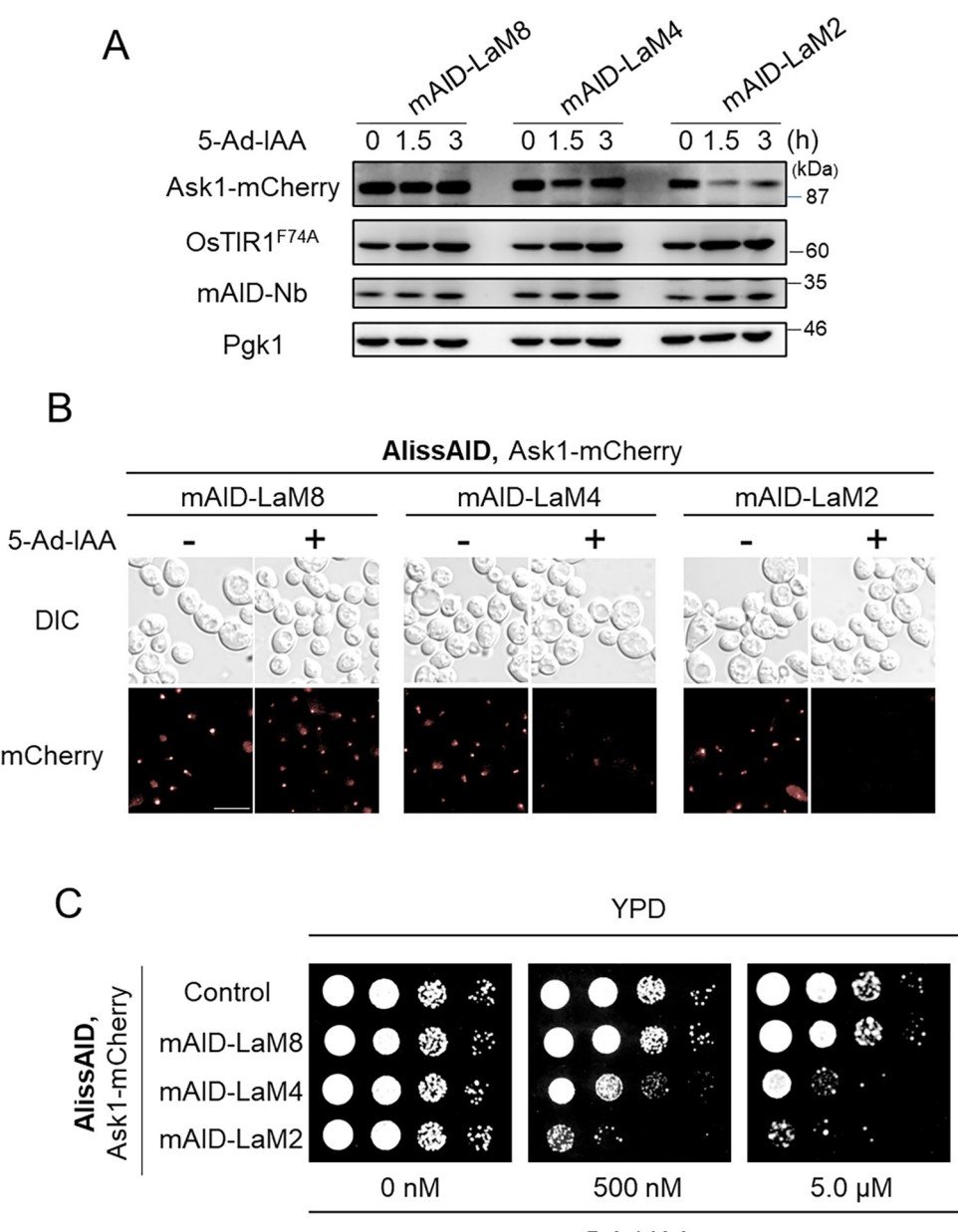

**Fig 6. mCherry fusion protein is degraded effectively using LaM2 and LaM4.** (A) Immunoblotting of endogenous tagged Ask1-mCherry. Cells expressing mAID-LaM8, mAID-LaM4, and mAID-LaM2 were established from the same cell line expressing OsTIR1$^{F74A}$ and endogenous fusion Ask1-mCherry. (B) Fluorescence of Ask1-mCherry. Cells were treated with 5.0 μM 5-Ad-IAA for 3 h. Scale bar, 5 μm. (C) Serial dilution spotting on the YPD medium containing 5-Ad-IAA. Cells were grown for 24 h at 30˚C.

IAA, Ask1-mCherry fluorescence signals were diminished in mAID-LaM2- and mAID-La-M4-expressing cells but not in mAID-LaM8-expressing cells (Figs 6B and S5C). The mAI-D-LaM2 expressing cell line showed severe growth defects in the presence of 5-Ad-IAA in serial dilution spotting assay. While the mAID-LaM4 expressing cell line showed mild growth defects, the mAID-LaM8 expressing cell line did not show any growth defects (Fig 6C). A similar trend was also observed in degron strains targeting Neo1-mCherry-Flag (S5D Fig). These

results indicate that LaM2 is a good candidate for the AlissAID system for mCherry fusion proteins in budding yeast.

## Discussion

Methods to induce target-protein degradation in cells often negatively affect the cells themselves. Inactivation of target proteins by temperature-sensitive mutants requires a temperature shift from 23–28°C to 37°C [34]. Such severe temperature changes have significant effects on organisms. Furthermore, in conventional AID systems, target-protein degradation requires high concentrations of auxins, which negatively affects organisms. Our AlissAID system requires only 50 nM of 5-Ad-IAA for target-protein degradation, potentially preventing this problem.

In this study, we found that the AlissAID system shows milder basal degradation than the ssAID system. It might be due to their different methods for the interaction between $OsTIR1^{F74A}$ and the target protein: AlissAID system uses an indirect interaction via mAID-Nb, while ssAID system uses a direct interaction. Therefore, AlissAID system would be a good inducible degron system for the proteins that will be destabilized by AID tagging.

Not all nanobodies (single-peptide antibody) work well in our AlissAID system. We tested three different anti-mCherry nanobodies (LaM2, LaM4, and LaM8). Among them, LaM2 induced target protein degradation most efficiently, whereas LaM4 weakly induced target degradation. In terms of mCherry-binding forces, LaM2 and LaM4 have KD values of 3.02 ± 1.88 nM and 22.50 ± 34.6 nM, respectively [35]. Considering that the GFP-banding force of VHHGFP4 has a KD value of 1.40 nM [36], degradation efficiency appears to be associated with the affinity to the nanobody and target proteins. In terms of crystal structures, the mCherry LaM2 binding site is similar to the GFP binding site, which VHHGFP4 binds to [35,36]. These binding sites may potentially be related to efficient ubiquitination and degradation. In conclusion, target-protein degradation efficiency is largely dependent on the type of nanobodies that are used in the AlissAID system.

In conclusion, this study demonstrates that GFP or mCherry fusion proteins can be degraded by the AlissAID system in budding yeast. In this study, we used nanobodies, which recognized GFP or mCherry proteins for degradation. Other binding domains, including monobodies [37] [38], DARPins [39] or binding peptides [40] that bind specific proteins can also be potentially used. The use of nanobodies, which recognize target protein directly in cells, would also enable degradation of untagged endogenous proteins using the AlissAID system. In budding yeast, there is a large clone collection of GFP-tagged strains. This clone collection enables the generation of various types of AlissAID strains systematically. A combination of the GFP clone collection and robotics would be useful for high-throughput screening and genome-wide analysis in budding yeast. Similar to the conventional AID system, we expect the AlissAID system to function in various eukaryotic organisms. This new method provides an ideal protein-knockdown system for a wide range of target proteins using various binding domains, including nanobodies.

## Materials and methods

### Yeast strains and media

The *Saccharomyces cerevisiae* strains used in this work are listed in S1 Table. Cells were grown in YPD (2% D-glucose, 1% yeast extract, and 2% peptone,) supplemented with 100 mg/L adenine or SD medium (2% D-glucose and 0.67% yeast nitrogen base without amino acids) supplemented with appropriate amino acids, uracil, tryptophan, histidine, and leucine.

## Auxin derivative stock solutions

3-Indoleacetic Acid (IAA) (Nacalai tesque: 19119–61) and 5-Adamantyl-IAA (Tokyo Chemical Industry: A3390) were diluted into dimethyl sulfoxide (DMSO) at 500 mM or 5 mM concentration and stored at -30˚C. When mixed with plate medium, a solution of 1000 times the concentration of each final concentration was prepared, and the medium was prepared by adding 1/1000 to the medium.

## Genetic manipulation

Chromosome fusions of *GFP*, *mAID-GFP* and *mCherry* to the 3′-terminus of the gene were conducted using PCR-based gene modification [41]. The sequence containing the tag [*GFP*, *mAID-GFP*, *mCherry*], the *ADH1* terminator, and a marker gene was amplified using PCR from pFA6a GFP-His3MX6 [for GFP], pFA6a mAID-GFP-His3MX [for mAID-GFP](pYS15, this study), pFA6a mCherry-His3MX6[for mCherry](our laboratory) with a primer set containing the homologous region(50–55 mer)of each gene. The PCR-amplified fragments were directly inserted into the chromosome via homologous recombination. Successful deletions of the genes and tagging were confirmed via genomic PCR, immunoblot, and/or fluorescence microscopy. The sequence encoding *OsTIR1(F74A)*, *OsTIR1(F74G)*, *mAID(KR)-VHHGFP4 (KR)*, *mAID(KR)-LaM2(KR)*, *mAID(KR)-LaM4(KR)*, *mAID(KR)-LaM8(KR)* is integrated into *URA3* or *LEU2* locus as follows. The *ADH1* promoter sequence followed by *OsTIR1(F74A)*, *OsTIR1(F74G)*, *mAID(KR)-VHHGFP4(KR)*, *mAID(KR)-LaM2(KR)*, *mAID(KR)-LaM4(KR)*, *mAID(KR)-LaM8(KR)* sequence was constructed on pRS306 or pRS305 [32]. The resultant plasmids were linearized by digestion with *Stu*1 or *Bst*X1 or *Afl*2 and integrated into *URA3* or *LEU2* locus, respectively, via homologous recombination. The sequence encoding *OsTIR1 (WT)-T2A-mAID(KR)-VHHGFP4(KR)*, *OsTIR1(F74A)-T2A-mAID(KR)-VHHGFP4(KR)*, *OsTIR1(F74S)-T2A-mAID(KR)-VHHGFP4(KR)*, *OsTIR1(F74C)-T2A-mAID(KR)-VHHGFP4 (KR)* is integrated into *URA3* locus as follows. The *ADH1* promoter sequence followed by *OsTIR1(WT)-T2A-mAID(KR)-VHHGFP4(KR)*, *OsTIR1(F74A)-T2A-mAID(KR)-VHHGFP4 (KR)*, *OsTIR1(F74S)-T2A-mAID(KR)-VHHGFP4(KR)*, *OsTIR1(F74C)-T2A-mAID(KR)- VHHGFP4(KR)* was constructed on pRS306. And BY4741 *URA3* locus sequence is constructed between *Eco*R1 and *Xho*1 site on same plasmids. The resultant plasmids were linearized by digestion with *Sma*1 and integrated into the *URA3* locus, respectively, via homologous recombination.

## Immunoblot analysis

Cell lysates were prepared using the alkaline-trichloroacetic acid method. Harvested cells were resuspended in an ice-cold alkaline solution containing 0.25 M of NaOH and 1% (v/v) of 2-mercaptoethanol. After incubation on ice for 10 min, Trichloroacetic acid was added at a final concentration of 7% (w/v). After centrifugation at $20,000 \times g$ for 2 min at 4˚C, the supernatant was removed. Then, 1 M tris was gently added, ensuring to not break the pellet, to neutralize the remaining trichloroacetic acid, and centrifuged at $20,000 \times g$ for 30 s. After the removal of supernatant, proteins were eluted by incubation in SDS-sample buffer for 5 min at 95˚C. After centrifugation at $10,000 \times g$ for 1 min at room temperature, the supernatant was transferred to a new tube and subjected to immunoblotting. Proteins were separated via SDS-PAGE and transferred to a polyvinylidene difluoride membrane (Millipore, Billerica, USA). The membrane was incubated with anti-GFP (1:2000 dilution; our laboratory), anti-RFP(1:2000 dilution; Chromotek, 5f8-20/5f8-100), anti-Pgk1 (1:5000 dilution; our laboratory), anti-TIR1 (1:2000 dilution; Medical and Biological Laboratories, PD048), anti-AID (1:2000 dilution; gifted from Prof. Karim Labib), anti-Mcm4 (1:1000 dilution; gifted from Prof. Karim Labib). HRP-conjugated

anti-Rabbit IgG (1:5000 dilution; Sigma, A6154) or Peroxidase-conjugated anti-Sheep IgG (1:5000 dilution; 713-035-003, Jackson ImmnoResearch) or Peroxidase-conjugated anti-Rat IgG (1:5000 dilution; 112-035-003, Jackson ImmnoResearch) was used as the secondary antibody. Immunodetection was performed using a Luminata Forte Western HRP Substrate system (Merck Millipore, Burlington, USA, 61-0206-81) or a Chemi-Lumi One L system (Nacalai Tesque, 07880) with a bioanalyzer (LAS4000 mini; GE Healthcare Biosciences).

## Fluorescence microscopy

Cells were grown to log phase in SD medium containing 5 μM 5-Ad-IAA for 3 hours at 30˚C. After centrifugation, cells were fixed by adding 4% formaldehyde in PBS for 10 min at RT. Cells were collected by centrifugation. After washing with PBS, cells were resuspended in PBS and observed under a fluorescence microscope (AxioObserver Z1; Carl Zeiss, Oberkochen, Germany) equipped with a CCD camera (AxioCam MRm; Carl Zeiss). Exposure time GFP:2,000 ms, mCherry:4,000 ms. Fluorescence intensity was analyzed using ImageJ for photographs taken under the same conditions.

## Spot assay (serial dilution assay)

Yeast cells were grown in YPD or SD medium overnight at 30˚C until $OD_{600}$ reached 0.7–0.8. Cells were collected and suspended in 200 μL of sterile water (0.3 $OD_{600}$ equivalent). Ten-fold serial dilutions were generated as follows. 200 μL of the cell suspension was transferred to the first lane of a 96-well plate. 180 μL of sterile water were placed in the second, third, and fourth lanes. Then, 20 μL of the cell suspension in the first lane was transferred to the second lane (lane 2) and mixed by pipetting ten times. Similarly, 20 μL of the cell suspension was transferred from the second lane to the third lane, and from the third lane to the fourth lane, as described above. An aliquot of the cell suspension (3 μL on YPD plate, 5 μL on SD plate) was spotted on agar plates containing IAA, 5-Ad-IAA, or G418, which were then incubated at 30˚C.

## Flow cytometry

Cells were grown in YPD medium overnight at 25˚C not to exceed $OD_{600} = 1$. After treated with 5 μM 5-Ad-IAA at 30˚C, cells were centrifugated (3,000 g, 5 min) to pellet and discard the medium. Cells were resuspended and fixed in 70% ethanol at -30˚C for 1Day. Fixed cell were centrifugated, treated with 100 μl stain mix (PBS, 1 mg/ml RNase, 1/1000 SYBR Green I Nucleic Acid Gel Stain, Thermo Fisher Scientific) at 37˚C for 30 min. Add 900 μl PBS, vortex, cells were acquired by Attune Flow Cytometer (Thermo Fisher Scientific).

## Statistical analysis

Immunoblot signals were quantified with Image Quant TL (GE Healthcare Biosciences) and then normalized by using Pgk1 as a loading control. Error bars indicate standard deviations (three biological replicates). The *p*-values were determined using an unpaired Student's *t*-test. n.s., nonsignificant. In cases where multiple target bands were appeared, all the bands were quantified.

## Plasmid deposition

These plasmids are available from Addgene. pRS305-mAID-Nb (VHHGFP4): #198412, pRS305-mAID-LaM2: #198413, pRS305-mAID-LaM4: #198414, pRS305-mAID-LaM8:

#198415, pRS306-OsTIR1$^{F74A}$: #198416, pRS316-OsTIR1$^{F74A}$-T2A-mAID-Nb (VHHGFP4): #198417

## Supporting information

**S1 Fig. OsTIR1$^{F74A}$ is better than OsTIR1$^{F74G}$ in the AlissAID system.** (A) Immunoblots of AlissAID strains that had OsTIR1 (OsTIR1 F74A or OsTIR1 F74G), showing the 5-Ad-IAA-triggered degradation of a target protein (Neo1-GFP). CBB-stained proteins were used as a loading control. (B) Serial dilution spotting assay of AlissAID strains on YPD medium. Each strain that had an essential protein (Ask1-GFP or Neo1-GFP) as a target protein was grown with or without 5-Ad-IAA at 30˚C for 24 h.
(PDF)

**S2 Fig. Degradation of GFP-fusion proteins and phenotyping by the AlissAID system.** (A) Immunoblot of WT and Mcm4-AlissAID strain by the anti-Mcm4 antibody. Cells were treated with 5μM 5-Ad-IAA for 1h. Pgk1 was used as a loading control. (B) Immunoblots of AlissAID strains toward Mcm4, Ask1 and Neo1. Cells were treated with 5 μM 5-Ad-IAA for 0, 1, 2, and 3 h. (C) Expression levels of Mcm4, Ask1 and Neo1 proteins in the absence of 5-Ad-IAA. Signal intensities of Mcm4-GFP, Ask1-GFP and Neo1-GFP bands were normalized by using the loading control Pgk1. (D) Degradation profiles of Mcm4, Ask1 and Neo1 in AlissAID system. Signal intensities of Mcm4-GFP, Ask1-GFP and Neo1- GFP bands were normalized by using the loading control Pgk1. Relative levels of these proteins at time zero are shown. (E) Cell cycle profiles of Mcm4- and Ask1-AlissAID strains. Cells were treated with 5 μM 5-Ad-IAA for 0, 1 and 2 h.
(PDF)

**S3 Fig. AlissAID system resolves basal degradation issue.** Comparison of the levels of a target protein (Mcm4-GFP or ASK-GFP) in AlissAID strain with those in the control strain that had the respective target protein, but neither OsTIR1 nor mAID-Nb. After normalizing signal intensities on the immunoblot with the loading control Pgk1, target protein levels were statistically analyzed. Means ± SD (n = 3 biological replicates); n.s., nonsignificant; unpaired Student's *t*-test.
(PDF)

**S4 Fig. Target protein degradation in simplified prepared degron strain.** (A) Fluorescence microscopy observations of AlissAID strains toward Mcm4, Cdc45, Ask1, Neo1, Las1, Rrn3 and Apc4. These AlissAID strains were generated from GFP clone collection. Cells were treated with or without 5.0 μM 5-Ad-IAA for 3 h. Scale bar, 5 μm. (B) Schematic diagram of subcellular localization of Neo1, Dpm1 and Sam35. (C) Immunoblots of Dpm1- and Sam35-AlissAID strain. Cells were treated with 5 μM 5-Ad-IAA for 1h. Pgk1 was used as a loading control. (D) Serial dilution spotting of the control, Neo1-, Dpm1- and Sam35-AlissAID strains on YPD medium with or without 5 μM 5-Ad-IAA. Cells were grown for 24 h at 30˚C.
(PDF)

**S5 Fig. AlissAID system is available for degradation of mCherry-tagged target proteins.** (A) Schematic illustration for generation of Ask1-AlissAID stains, which express mAID-LaMs (anti-mCherry nanobody). All LaM-expressing cell lines were established from mCherry tagged cell lines to the same OsTIR1$^{F74A}$-expressing line. (B) Immunoblots of three AlissAID strains, each of which had mAID-Nb (LaM2, LaM4, or LaM8) and a target protein (Ask1-mCherry). Pgk1 was used as a loading control. (C) Fluorescence intensities of Ask1-mCherry in

three independent AliceAID strains that were treated with 5-Ad-IAA for 0 and 3 h. Fluorescence intensities were acquired from the images in Fig 6B by Image and were normalized with the number of cells in the images. (D) Serial dilution spotting assay of AliceAID strains on YPD medium. Each strain that had mAID-Nb (LaM2, LaM4, or LaM8) and an mCherry-tagged essential protein (Neo1-mCherry-Flag) as a target protein was grown with 5-Ad-IAA at 30˚C for 24 h.
(PDF)

**S1 Table. Yeast strain list in this study.**
(PDF)

**S1 Data. Raw numerical data underlying graphs.**
(XLSX)

## Acknowledgments

We are grateful to Kunio Nakatsukasa (Nagoya City University) for providing GFP tagged budding yeast strains; to Ryo Fujisawa, Fabrizio Villa and Karim Labib (University of Dundee) for providing antibodies; to Keiko Kamura (Nagoya University) for her technical support.

## Author Contributions

**Conceptualization:** Yoshitaka Ogawa, Kohei Nishimura.

**Data curation:** Yoshitaka Ogawa, Keisuke Obara.

**Funding acquisition:** Kohei Nishimura, Takumi Kamura.

**Investigation:** Yoshitaka Ogawa, Keisuke Obara.

**Methodology:** Kohei Nishimura, Keisuke Obara.

**Project administration:** Kohei Nishimura, Takumi Kamura.

**Resources:** Keisuke Obara.

**Supervision:** Takumi Kamura.

**Validation:** Yoshitaka Ogawa.

**Writing – original draft:** Yoshitaka Ogawa, Kohei Nishimura.

**Writing – review & editing:** Kohei Nishimura, Keisuke Obara, Takumi Kamura.

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
