## [Decision Letter · Decision Letter 0]

31 Jan 2023

Dear Dr Nishimura,

Thank you very much for submitting your Research Article entitled 'Development of AlissAID system targeting GFP or mCherry fusion protein' to PLOS Genetics.

The manuscript was fully evaluated at the editorial level and by independent peer reviewers. The reviewers appreciated the attention to an important topic but identified some concerns that we ask you address in a revised manuscript.

We therefore ask you to modify the manuscript according to the review recommendations. Your revisions should address the specific points made by each reviewer.

Yours sincerely,

Aimee M. Dudley, Ph.D.

Academic Editor

PLOS Genetics

Quanjiang Ji

Section Editor

PLOS Genetics

All of the reviewers felt that the method, which builds up and improves previous methods, will be a valuable tool to the yeast community and may also aid the development of comparable methods in other organisms. However, the reviewers did identify several areas which would need to be addressed, in some cases by the inclusion of additional control experiments before the manuscript could be reconsidered at PLoS Genetics. In particular, I felt that confirming that the target protein itself (as opposed to just GFP) was degraded, either by the use of antibodies to either the native protein or an N-terminal tag (as suggested by Reviewer #1), would be important. Additionally, I am interested your thoughts on the suggestion by Reviewer #3 to examine the degradation of proteins in different subcellular compartments. Do we expect this system to be able to degrade proteins that have already been transported to a compartment such as the mitochondrial matrix prior to auxin induction? Or do we expect the system to be able to degrade newly translated polypeptides that are en route to the mitochondria post-induction? I would be happy to discuss this point via email.

Reviewer's Responses to Questions

**Comments to the Authors:**

Reviewer #1: Summary:

In this study, Ogawa and Nishimura et al. developed an affinity linker-based super sensitive auxin-inducible degron (AlissAID) by using a single domain antibody (nanobody) in a budding yeast. This system can degrade the desired target proteins by administering a synthetic auxin, namely, 5-adamantyl-IAA (5-Ad-IAA), at nanomolar concentrations. To prove the usefulness of this system, the authors utilized it for the degradation of several essential proteins. They also developed the “AlissAID plasmid,” a single plasmid that encodes both OS TIR1 (F74A) and mAID (KR)-nanobody (KR) and may be used to systematically degrade GFP-tagged proteins. Finally, the authors demonstrated that this system could be applied to eliminate mCherry-fused proteins by using the nanobody that specifically recognizes mCherry. Overall, the AlissAID system would be a highly efficient tool for the genetic analysis of budding yeasts and other higher eukaryotes.

Major comments:

The authors assessed the degradation of GFP-fused target proteins by western blotting with the GFP antibody and confirmed it by analyzing cell growth and cell cycle progression. However, these analyses are indirect. It is formally possible that only the GFP moiety was degraded, and a portion of target proteins likely remained intact. Therefore, the authors should demonstrate the extent of target protein degradation by using an endogenous antibody or other epitope tags at the N-terminus (e.g., HA, myc, or FLAG). This aspect is important because this study is the first to characterize the novel protein knockdown system.

Minor comments:

1. In Fig. 2A, the mAID-Nb level increased after 5-Ad-IAA was administered in Mcm4-GFP cells and Ask1-GFP cells but not in Neo1-GFP cells. Please provide an explanation for these results.

2. In Fig. 2C, the cells expressing OSTIR1 (F74A) and mAID-Nb seem to grow better than the control cells do. Is there any reason?

3. In Fig. 3C, “hour” may be removed from “5-Ad-(IAA)(hour).”

4. In Fig. S1, the authors need to show the FACS data of 5-AD-IAA-treated cells that do not express OSTIR1F74A and mAID-Nb. In addition, the profile of the DNA content of Mcm4-GFP cells at 0 h likely differs from that of Ask1-GFP cells. Does the attachment of GFP to Ask1 affect the cell cycle?

5. The authors tested this system only at 30 °C. Can this system be used at other (low and high) temperatures?

6. The methods of statistical analyses should be presented more clearly in both the Methods section and Figure legends. For example, in Fig. 3 and 5, do error bars stand for standard deviation or standard error? How were the protein levels normalized? How was the p-value calculated? What was the statistical method used?

7. The molecular weight (MW) is shown in some blots only. The authors should show the MW markers in all blots.

Reviewer #2: The authors of this manuscript have developed a new variant of the auxin-inducible degradation (AID) system for use with GFP- or mCherry-tagged proteins in budding yeast. What is of particular interest here is that this work combines two AID advances into a single system in budding yeast: (1) the super sensitive AID (ssAID) that requires only minute amounts of synthetic auxin and thus reduces inducer toxicity and target basal degradation issues, and (2) a GFP-targeting nanobody fused to an AID-tag that enables any GFP-tagged protein to be targeted for inducible degradation, minimizing the need to pre-tag targets with the AID tag. The authors demonstrate that this AlissAID system works in budding yeast and is an improvement from previous AID systems. They confirmed via a variety of biochemical approaches that AlissAID worked against several different yeast target proteins and that the two component system could be introduced via a single plasmid transformation. Finally, they demonstrated that this approach can move beyond GFP-tagged cells, showing that mCherry nanobodies can also be successfully introduced into the AlissAID system.

Since many proteins of interest have already been GFP-tagged in cell lines, this improves the ease of use of the AID targeted degradation system since it does not require the additional step of AID-tagging individual target proteins. AlissAID is a valuable tool, especially for yeast biologists who have access to the large Yeast GFP Clone Collection strain library. AlissAID is likely to be of particular interest for medium-to-high throughput screening approaches that can make use of this library and the single-step AlissAID transformation process. The successful expansion of AlissAID to target mCherry-fusion proteins also suggests that any protein for which a tight-binding nanobody exists could make use of this tool for inducible-degradation studies. While AlissAID is a valuable advance for the field of AID development and especially the yeast geneticist’s toolbox, it does not necessarily represent a large leap forward from the previous AID systems from which it was built.

I really enjoyed reading this well-crafted manuscript featuring a new AID advance and I look forward to seeing AlissAID becoming more widely used within the yeast biology community and beyond!

Introduction

Provided a succinct yet thorough background on past AID advances and limitations, and the logical next step of merging the ssAID and AID-tagged nanobodies approaches into a single approach.

Line 73:” that OsTIR1F74A showed a mild sensitivity to IAA” could the authors clarify precisely what type of sensitivity they are referring to here? It was a little unclear since the previous sentence referred to basal degradation but this is clearly talking about in the presence of IAA.

Results

Overall, I thought the data and analyses supported the claims made by the authors, and the Results section generally proceeded logically. Below, I have noted a couple places where either additional rationale or changing section order could help readers follow the authors’ logic more clearly.

Fig 1: is a useful visual to understand the differences between the various AID systems referred to in this work, and could actually be cited when they come up in the Introduction too. Please include in 1C a label for the nanobody component, and consider shortening “AID-tag(IAA17)” to “mAID” for simplicity?

Fig 2: in 2C the (-) sign was not obvious to me at first, it could be made a bit bigger or bolder and then rather than the “OsTIR1F74A, mAID-nb” label that takes up two lines, simplify to “AlissAID” so that it will fit on a single line. I suggest that in this and all other figures, the authors consider using the simplified label “AlissAID” when possible to decrease the amount of text on the figures. In S1 referred to at line 132, it actually looks like very little effect at 1 hr for Mcm4, but at later time points both Mcm4 and Ask1 actually show G2/M arrest?

Fig 3: Lines 143-145 This rationale feels like it comes out of nowhere describing a key piece of data before showing that data…maybe the authors could reframe it by rearranging to lead with the sentence “There are questions about..”, followed by “To approach this…” and then “We coincidentally found…”

Again, the authors could simplify the labels to AlissAid and ssAID.

Line 152 - before this sentence, flow could be improved by explicitly stating rationale here, for example something like “To confirm that this is due to basal degradation due to presence of OsTIR1, we also built strains lacking receptors…”

In 3C, there are two Neo1-GFP bands in each lane in each set of strains - which one is being quantified? The Neo1-GFP bands are hard to resolve in the Neo1-GFP strains, and expression does not match above in A at 0 inducer (i.e. expression is very strong in the blots in A compared to the Neo1-mAID-GFP, but the reverse seems to be true in the blots in C?). Did the blots not shown from the other two independent experiments show this or a different trend? Is there some degree of variability in expression level?

Fig 4: At Line 558, the legend could use a bit more description of the 4A cartoon: what is depicted in top vs bottom (i.e. multistep vs singlestep process).

Fig 5: Maybe this Figure and section (Lines 184-197) would make more sense right after Fig 1? Since most of the remainder of the paper uses the F74A mutation, this provides rationale for that choice throughout.

Fig 6: At Line 207, maybe briefly mention the difference between LaMs 2,4,and 8 to provide some rationale for why all three were tested?

In S4C, please clarify in the legend (Lines 524-526) that this is the quantification of the data from Fig 6b? Also, it looks like labels might be mixed up as Lam2 shows no degradation but microscopy image in 6b shows strong degradation. In S4D, the Lam2 data does not match data in C?

Discussion

With the exception of the comment below, I found the Discussion to be straightforward and especially appreciated the exciting possibilities for future applications laid in the last paragraph. I found the second paragraph (Lines 230-245) a bit challenging to follow and had to re-read several times to try to make sense of what was being said. Some of the sentences are fragmentary, and I think this paragraph would benefit from some language editing. At 234, I think this is citing a different paper?

Methods

Very detailed and straightforward. Will the plasmids be made available through a repository such as Addgene?

Miscellaneous

The paper is generally well-written but I do recommend some language editing for minor grammar and sentence structure revisions.

Reviewer #3: In this manuscript, authors propose to combine the available collection of yeast fluorescent protein (GFP and mCherry)-tagged proteins with the auxin-inducible degron (AID) system. They provide evidence that co-expression of AID-anti-FP-nanobody with a modified OsTIR1 adaptor will stimulate the auxin (5-Ad-IAA)-dependent degradation of several FP-tagged yeast proteins. The authors also argued that co-expression of AID-anti-FP-nanobody with FP-tagged protein reduces “basal” degradation of FP-tagged proteins in the absence of 5-Ad-IAA. While this novel combination of well-established tools could simplify the acute depletion of endogenously tagged yeast proteins, the manuscript will benefit from additional controls and a more accurate time course of AID-dependent degradation.

Specific critique:

1. Yeasts are divided every 60-90 minutes; therefore, 1h degradation could be too slow compared to the speed of cellular processes in the yeast cell. It will be essential to test a more detailed (10, 20, 30, 45 min) time course of degradation presented in Figure 1.

2. To make the AlissAID system more universal, it will be important to compare the time course of degradation for both high- and low-expressed yeast proteins.

It will also be essential to compare the degradation of FP-tagged protein localized in different compartments (cytosol, nucleus, peripheral and integral components of ER, Golgi, mitochondria, peroxisomes, vacuoles and plasma membrane).

3. The size of mAID (7.4 kDa) tag is much smaller than GFP (26 kDA) and it will be more appropriate to compare the properties of ORF-FP vs ORF-mAID rather than use the awkward ORF-mAID-FP hybrid.

4. Since the nanobody binding to FP protein is very tight, it is not clear why the basal degradation of ORF-mAID-FP hybrid will be higher as compared to ORF-FP coexpressed with AlissAID system.

**Have all data underlying the figures and results presented in the manuscript been provided?**

Reviewer #1: Yes

Reviewer #2: Yes

Reviewer #3: Yes

PLOS authors have the option to publish the peer review history of their article (what does this mean?). If published, this will include your full peer review and any attached files.

Reviewer #1: No

Reviewer #2: No

Reviewer #3: No

---

## [Decision Letter · Decision Letter 1]

4 Apr 2023

Dear Dr Nishimura,

We are pleased to inform you that your manuscript entitled "Development of AlissAID system targeting GFP or mCherry fusion protein" has been editorially accepted for publication in PLOS Genetics. Congratulations!

Yours sincerely,

Aimee M. Dudley, Ph.D.

Academic Editor

PLOS Genetics

Quanjiang Ji

Section Editor

PLOS Genetics

Comments from the reviewers (if applicable):

Both reviewers agreed that you have addressed their original concerns, with one exception. Reviewer #3 requested that a statement be added to the abstract that helps readers understand which cellular compartments the method is expected to work in. Text similar to that included in lines 213-215 of revised manuscript would be acceptable, e.g. "Our results indicate that, similar to the conventional AID system, antigen recognition sites must be exposed in cytosol or nucleus to be degraded by the AlissAID system." Assuming that you agree, please include the revised abstract in your final manuscript submission. If you have and questions or concerns, feel free to contact me directly or via the journal staff.

Reviewer's Responses to Questions

**Comments to the Authors:**

Reviewer #1: The authors have adequately addressed the comments made by the reviewers in the

revised version of the manuscript. I have no further comments.

Reviewer #3: The authors addressed most of my critiques effectively, except for one crucial aspect. In my previous communication, I requested the authors to compare the degradation rates of FP-tagged proteins in different cellular compartments, including the cytosol, nucleus, and various organelles such as the ER, Golgi, mitochondria, peroxisomes, vacuoles, and plasma membrane. However, the authors only compared the degradation rates of two FP-tagged TM proteins with different topologies (cytoplasmic versus luminal), which is not sufficient to answer my question. This is because the luminal FP tag cannot be accessed by cytoplasmically expressed FP-nanobody and Tir1. As this is a methods paper, it is essential to systematically test the degradation potential of peripheral and integral membrane proteins with cytoplasmically exposed FP tags, for the proposed methodology to be widely applicable. Without this crucial experiment, the proposed methodology is limited to cytoplasmic and mitochondrial proteins, and this limitation should be clearly stated in the abstract.

**Have all data underlying the figures and results presented in the manuscript been provided?**

Reviewer #1: None

Reviewer #3: None

PLOS authors have the option to publish the peer review history of their article (what does this mean?). If published, this will include your full peer review and any attached files.

Reviewer #1: No

Reviewer #3: No

**Data Deposition**

http://datadryad.org/submit?journalID=pgenetics&manu=PGENETICS-D-22-01432R1

**Press Queries**

---

## [Editor Report · Acceptance letter]

12 May 2023

PGENETICS-D-22-01432R1 

Development of AlissAID system targeting GFP or mCherry fusion protein 

Dear Dr Nishimura, 

We are pleased to inform you that your manuscript entitled "Development of AlissAID system targeting GFP or mCherry fusion protein" has been formally accepted for publication in PLOS Genetics! Your manuscript is now with our production department and you will be notified of the publication date in due course.

With kind regards,

Timea Kemeri-Szekernyes

PLOS Genetics

On behalf of:
